# Multiple clades of Husavirus in South America revealed by next generation sequencing

**Endrya do Socorro Fôro Ramos**[1], **Ulisses Alves Rosa**[1], **Geovani de Oliveira Ribeiro**[1], **Fabiola Villanova**[1], **Flávio Augusto de Pádua Milagres**[2,3], **Rafael Brustulin**[2,3], **Vanessa dos Santos Morais**[4], **Emerson Luiz Lima Araújo**[5], **Ramendra Pati Pandey**[6], **V. Samuel Raj**[6], **Ester Cerdeira Sabino**[4], **Xutao Deng**[7,8], **Eric Delwart**[7,8], **Adriana Luchs**[9], **Élcio Leal**[1]\*, **Antonio Charlys da Costa**[4]\*

**1** Laboratório de Diversidade Viral, Instituto de Ciências Biológicas, Universidade Federal do Pará, Belém, Pará, Brazil, **2** Secretary of Health of Tocantins, Tocantins, Brazil, **3** Public Health Laboratory of Tocantins State (LACEN/TO), Tocantins, Brazil, **4** Departamento de Moléstias Infecciosas e Parasitárias, Instituto de Medicina Tropical da Faculdade de Medicina da Universidade de São Paulo, São Paulo, Brazil, **5** General Coordination of Public Health Laboratories of the Strategic Articulation, Department of the Health Surveillance Secretariat of the Ministry of Health (CGLAB/DAEVS/SVS-MS), Brasília, Federal District, Brazil, **6** Centre for Drug Design Discovery and Development (C4D), SRM University, Delhi-NCR, Rajiv Gandhi Education City, Sonepat, Haryana, India, **7** Vitalant Research Institute, San Francisco, California, United States of America, **8** Department Laboratory Medicine, University of California San Francisco, San Francisco, California, United States of America, **9** Enteric Diseases Laboratory, Virology Center, Adolfo Lutz Institute, São Paulo, Brazil

\* charlysbr@yahoo.com.br (ACdC); elcioleal@gmail.com (ÉL)

**Data Availability Statement:** Sequences generated in this study are available in Genbank (MT798545-MT798549).

**Funding:** ACC has a scholarship provided by Fundacao de Amparo a Pesquisa do Estado de Sao

## Abstract

Husavirus (HuV) is an unclassified virus of the order *Picornavirales* that has already been identified worldwide in various locations. The genetic, epidemiological, and pathogenic characteristics are, however, little understood. In children with acute gastroenteritis, this study used next-generation sequencing to recognize unknown sources of viruses. In particular, 251 fecal samples obtained from individuals were sequenced in southern, northeastern, and northern Brazil. all samples were also analyzed using culture methods and parasitological tests to classify other enteric pathogens such as bacteria, parasites, and viruses. 1.9% of the samples tested positive for HuV, for a total of 5 positive children, with a mean age of 2 year, with three males and two females. Detailed molecular characterization of full genomes showed that Brazilian HuVs' nucleotide divergence is less than 11%. The genetic gap between Brazilian sequences and the closest HuV reported previously, on the other hand, is 18%. The study showed that Brazilian sequences are closely related to the HuV defined in Viet Nam in 2013, further characterization based on phylogenetics. At least two divergent clades of HuV in South America were also seen in the phylogenetic study.

## Introduction

The number of novel and divergent viruses in the order *Picornavirales* was significantly increased by next-generation DNA sequencing (NGS) [1–5]. *Caliciviridae*, *Dicistroviridae*, *Iflaviridae*, *Marnaviridae*, *Picornaviridae*, *Polycipiviridae*, *Secoviridae* and *Solinviviridae* [3, 4] are classified into eight recognized families in this order. Most viruses of this order are capable of

Paulo/ FAPESP (2017/00021-9), EL has a scholarship provided by the Conselho Nacional de Pesquisa/CNPq (302677/2019-4) and AL is supported by CNPq (400450/2016-0) and FAPESP (2015/12944-9). The funders had no role in study design, data collection and analysis, decision to publish, or preparation of the manuscript.

**Competing interests:** The authors have declared that no competing interests exist.

infecting a wide range of hosts, including invertebrates, vertebrates, plants, fungi, and algae, and of causing a variety of host clinical conditions, including the common cold, acute diarrhea, heart disease, liver disease, and extreme neuropathy [6–12].

Posa-like viruses (posavirus = porcine stool-associated RNA viruses), a group of *Picornavirales* consisting of divergent viruses that share a similar genome architecture, have been found predominantly in fecal samples in a wide range of hosts [4, 6, 8, 9, 13–18]. In pigs and water barns obtained from swine farms, posaviruses were detected by NGS [15, 16, 19], fish stool-associated RNA virus (fisavirus) was identified in the intestinal content of carps [17], and human stool-associated RNA virus (husavirus) was identified in primarily stable human feces [10]. Other members, such as panda-associated virus (pansavirus), bat stool-associated RNA virus (basavirus), and rat stool-associated RNA virus are also included in this community [15–21].

In particular, Husavirus (HuV) is similar to other viruses of the order *Picornavirales*, having a single-strand positive genome (+ ssRNA-monopartite), around 7.2 to 9.8 kb in length. Furthermore, all of its self-cleaved polyprotein constituents and a replication block consisting of three Hel-Pro-Pol domains (Helicase, Type 3C Protease, and RNA-RdRp-dependent Polymerase) are capable of encoding one or two polyproteins, although there are exceptions [2, 6]. In 2015, HuV was first identified in the Netherlands from human fecal specimens obtained from a group of patients in Amsterdam between 1984 and 2014 (HIV-1 positive and HIV-1 negative) [1]. It was later identified in Vietnam, Ethiopia, and Venezuela in various studies [2–4]. While HuV has already been identified in various continents (Europe, Africa, Asia, and America), its pathogenicity is still little understood. It is also possible that most Posa-like viruses found in animals are, or have a dietary or environmental cause, infecting gut commensal species [15, 18, 19].

Therefore the purpose of the present study is to report for the first time in Brazil on the identification and phylogenetic characterization of HuV in order to contribute to more knowledge on this new viral agent, because of the considerable constraints mainly on the pathogenicity, epidemiology, and molecular characteristics of HuV.

## Materials and methods

### Ethical aspects

The study was conducted in accordance with the Declaration of Helsinki of 1975 (https://www.wma.net/what-we-do/medical-ethics/declaration-of-helsinki/), revised in 2013. The protocol was approved by the Ethics Committee of the institutions involved (Faculdade de Medicina, Universidad de São Paulo (CAAE: 53153916.7.0000.0065), and Centro Universitario Luterano de Palmas—ULBRA (CAAE: 53153916.7.3007.5516), and all participants, the informed consent form was obtained, signed by the parents or guardians of the children involved in the study.

### Study population and specimen collection

The current cross-sectional surveillance study was carried out from 2010 to 2016 in the states of Tocantins, Maranhão and Pará, respectively in Central, Northern and North regions of Brazil. Fecal samples were collected in 38 different localities. A total of 251 specimens were collected, being 245 samples from state of Tocantins and 3 samples from the state of Pará. Three samples were obtained from border municipalities (Estreito and Carolina) located between the state of Tocantins and the state of Maranhão (Northeast region of Brazil. A total of 236 stool specimens were obtained from children aged 1–5 years, 3 stool specimens from children aged

8–15 years, and 7 stool specimens from adults aged 20–78 years with gastroenteritis symptoms. In five stool samples the age of the patient was missing.

This study was carried out with convenient surveillance specimens, without inclusion or exclusion criteria, and with no characterization of the participants; therefore, epidemiological data (*i.e.*, date of diarrhea onset, vomit episodes or fever) were not available for all patients.

## Sample screening

The samples described above were initially sent to the Public Health Laboratory of Tocantins (LACEN-TO), accompanied by a record of epidemiological investigation such as demographic data (age, sex, date of collection) and clinical data (signs and symptoms) of the participants. In this location, the identification of enteric pathogens such as bacteria (for example, *Escherichia coli* and *Salmonella* sp.) and parasites (for example, *Giardia* sp, *Taenia solium*) was carried out by means of culture techniques and conventional parasitological tests such as Hoffman's method and fresh direct examination. Samples were screened for viral enteric pathogens (*i.e.*, rotavirus and norovirus), using commercial enzyme immunoassays, such as RotaScreenII[®] and AdenoScreen[®]EIA (Microgen Bioproducts Ltd, 1, Watchmoor Point, Watchmoor Rd, Camberley GU15 3AD, UK). The samples were stored at -20 ˚C and the frozen fecal specimens were then taken to USP's Institute of Tropical Medicine (IMT / USP) to identify common enteric viruses (such as Rotavirus, Norovirus, Adenovirus, Astrovirus and Sapovirus) as well as rare or potential new viruses thought Next-generation Generation Sequencing (NGS) investigation. We also performed a PCR screening to determine the frequency of Husavirus in the individuals participants of this study. The nested PCR amplification was generally achieved by means of 35 cycles of denaturation at 95˚C for 50 s, annealing at 55˚C for 30 s and polymerization at 72˚C for 10 minutes, using the following primers (genome coordinates are indicate within parenthesis and are according to the sequence MG571863): 1outerF (5099–5122) 5'– CAACACTATGTGAAGAAGTTCCAG–3'; 1innerR (5387–5409) 5'–GGTGATGACCATGTGC TGTGTGT–3'; 1innerF (5121–5139) 5'– GGTGACTACAAGAACTTCG–3'; 1outerR (5497– 5476) 5'–CTTCTGGTCATCCGTGTAGATC–3'.

## Viral metagenomics

The protocol used to perform deep sequencing was a combination of several protocols applied to viral metagenomics and / or virus discovery according to the procedures previously described [22]. In summary, 50 mg of the human fecal sample was diluted in 500 µL of Hank's buffered saline (HBSS) and added to a 2 mL impact resistant tube containing C lysing matrix (MP Biomedicals, Santa Ana, CA, USA) and homogenized in a FastPrep-24 5G homogenizer (MP biomedicals, USA). The homogenized sample was centrifuged at 12,000 × g for 10 min and approximately 300 µL of the supernatant was then percolated through a 0.45 µm filter (Merck Millipore, Billerica, MA, USA) to remove bacterial and eukaryotic cells. Approximately 100 µL, PEG-it Virus Precipitation solution (System Biosciences, Palo Alto, CA, USA) was added to the filtrate and the contents of the tube were gently homogenized and then incubated at 4 ˚C for 24 h. After the incubation period, the mixture was centrifuged at 10,000 × g for 30 min at 4 ˚C and the supernatant (~ 350 µL) was discarded. The granulate, rich in viral particles, was treated with a combination of nuclease enzymes (TURBO DNase and RNase Cocktail Enzyme Mix-Thermo Fischer Scientific, Waltham, MA, USA; Baseline-ZERO DNase DNase-Epicenter, Madison, WI, USA; Benzonase -Darmstadt, Darmstadt, Germany and RQ1 DNase-Free DNase and RNase A Solution-Promega, Madison, WI, USA) to digest unprotected nucleic acids. The resulting mixture was subsequently incubated at 37 ˚C for 2 h. Then, the viral nucleic acids were extracted using a viral DNA / RNA kit ZR & ZR-96 (Zymo Research, Irvine, CA, USA), according to the

manufacturer's instructions. The synthesis of the cDNA was performed with an AMV reverse transcription reagent (Promega, Madison, WI, USA). A second strand cDNA synthesis was performed using a large DNA polymerase I fragment (Klenow) (Promega). Subsequently, a Nextera XT Sample Preparation Kit (Illumina, San Diego, CA, USA) was used to build a DNA library, which was identified using double barcodes. The library was then purified using the ProNex®️ size selective purification system (Promega, WI, USA). Following the ProNex®️ purification, the quantity of each sample was normalized to ensure an equal representation of the library with the combined samples using the ProNex®️ NGS Library Quant Kit (Promega, WI, USA). For size range, Pippin Prep (Sage Science, Inc.) was used to select a 300 bp tablet (range 200 to 400 bp), which excluded very short and long fragments from the library. Before the generation of the cluster, the libraries were quantified again by qPCR using the ProNex®️ NGS Library Quant Kit (Promega, WI, USA). The library was sequenced in depth using a Hi-Seq. 2500 sequencer (Illumina, CA, USA) with ends of 126 bp.

Bioinformatics analysis was performed according to the protocol previously described by Deng et al [23]. Contigs, including sequences of enteric viruses, fungi, bacteria, and others, sharing a percentage nucleotide identity of 95% or less, were assembled from the sequence readings obtained by de-assembly. The resulting singlets and contigs were analyzed using BLASTx to look for similarity with viral proteins on GenBank. The contigs were compared with GenBank's non-redundant nucleotide and protein databases (BLASTn and BLASTx). After virus identification, a reference model sequence was used to map the complete genome with the Geneious R9 software (Biomatters Ltd L2, 18 Shortland Street Auckland, 1010, New Zealand). Sequences generated in this study were deposited in the GenBank under the number: MT798545-MT798549.

## Alignments and annotation

Using Ugene software [24], the resulting contigs were subjected to updated protein blast searches to classify novel Picornavirales members. For further study, the genomes of HuV and other related viruses were selected based on the best results (best hits) from the BLASTx search. Using MAFFT tools, the next complete or nearly full genomes were aligned [25]. Using Gatu software [26], genome annotation was carried out and the HuV sequence KX673248 was used as a guide. The translated HuV sequences were used in the MotifFinder (https://www.genome.jp/tools/motif/) online server to evaluate viral motifs.

## Genetic identity

Genetic distance and their standard error were calculated using maximum composite likelihood model plus gamma correction and bootstrap with 100 replicates. Distances were calculated using MEGA software (Version X) [27]. To estimate the similarity of sequences we used a pair-wise method implemented in the program SDT [28]. To estimate the similarity alignments of every unique pair of sequences were done using algorithms implemented in MUSCLE [25]. After the computation of the identity score for each pair of sequences the program then uses the NEIGHBOR component of PHYLIP to compute a tree [29]. The rooted neighbour-joining phylogenetic tree orders all sequences according to their likely degrees of evolutionary relatedness. Results are presented in a frequency distribution of pairwise-identities in a graphical interface.

## Phylogenetic analysis

Phylogenetic trees were constructed using the maximum likelihood approach. To obtain reproducible results and provide greater reliability of clustering pattern of trees the statistical

support of branches was evaluated by approximate likelihood ratio test (aLRT). Trees were inferred using the FastTree [30] software and the GTR model plus gamma distribution and the proportion of invariable sites were used. Selection of the best model was done according to the likelihood ratio test (LRT) implemented in the jModeltest [31] software.

## Results

### Viral diversity

The most common agents found were rotavirus (RV) (44.62%), adenovirus (HdV) (33.47%), norovirus (NoV) (15.54%) and astrovirus (HAstV) and/or sapovirus (SaVs) from the 251 samples examined by laboratory tests or NSG (3.19% each). It is necessary to point out that after proper treatment, all patients were affected by acute diarrhea which resumed.

Concerning the patients in whom HuV was found, those characteristics have been summarized in Table 1. The presence of rotavirus group A, which is an important enteric pathogen and was not identified by other assay surveys, is an important finding for these patients. These patients have also been found to have other pathogenic viruses, such as noroviruses, sapoviruses, and adenoviruses (Table 1). Detailed description of viruses found in all 251 patients enrolled in this study was summarized in previous publications [32–40].

### Genome organization of Husavirus

Based on the best hits of BLASTx search we compared our sequences with HuV isolated from previous studies [1, 6, 7, 10–12, 15–19, 22]. Initially we performed a similarity analysis comparing Brazilian sequences with HuV described previously in a study performed with fecal samples [6, 12, 16]. We found that the nucleotide similarity in the polyprotein region of Brazilian sequences is 92%. The highest similarity of a Brazilian sequence and a reference HuV sequence (87%) was between TO-030 and KX673248 identified in 2013 in Viet Nam. The similarity of Brazilian sequences and other HuV sequences (MG571863, MG571864) detected in Amerindians from Venezuela is lower than 69% [12]. The overall similarity scores of all sequence pairs are shown in Fig 1.

To better explore the details of HuV genomes, we searched Pfam protein database and compared motifs detected in the sequence KX673248 with those detected in the sequence TO-030.

**Table 1. Characteristics of patients with HuV in the study.**

| ID* | DOB# | Gender | Collection date | Location | Symptoms | Other viruses identified by NGS |
|------|------|--------|-----------------|----------|----------|---------------------------------|
| TO-029 | 09/25/13 | Male | 11/25/14 | Xinguara/PA | Diarrhea, Vomiting | Rotavirus, Adenovirus, Parvovirus, Sapovirus |
| TO-030 | 10/20/11 | Female | 12/18/14 | Araguaina/TO | Diarrhea | Rotavirus, Norovirus |
| TO-067 | 04/03/13 | Female | 09/30/15 | Araguaina/TO | Diarrhea, Fever | Rotavirus, Adenovirus, Sapovirus, Norovirus, Gokushovirus, Enterovirus, Torque Teno Virus, Parechovirus |
| TO-127 | 05/15/13 | Male | 05/01/14 | Araguaina/TO | Diarrhea, Vomiting, Fever | Rotavirus, Adenovirus |
| TO-227 | 02/11/11 | Male | 08/31/14 | Pau D'Arco/TO | Diarrhea | Rotavirus, Adenovirus, Sapovirus |

*) Identification of patient,

#) Date of birth (Month/Day/year).

TO = Tocantins state; PA = Pará state

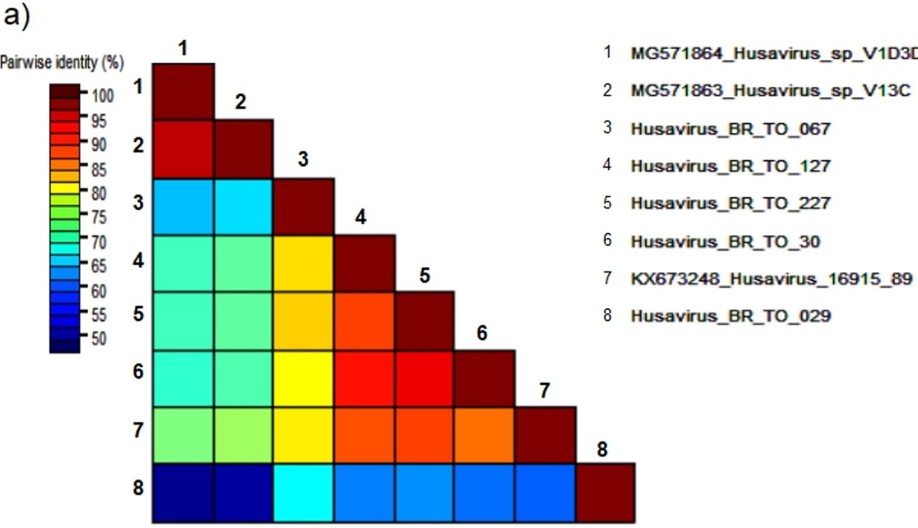

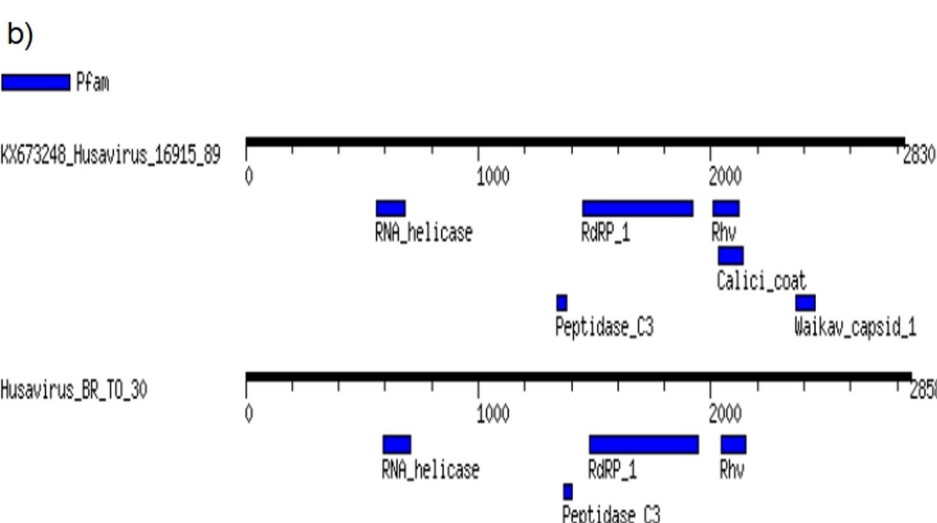

**Fig 1. Characteristics of HuV genomes.** (**a**) Nucleotide similarity matrix of HuV. The similarity of all pairs of sequences are indicated in colors according to the scale in the figure. (**b**) comparison between the polyprotein of KX673248 and TO-030. The diagram represents the genome of HuV (black line) and the main motifs detected by motif finder analysis (blue areas) using Pfam data base.

We found in the sequence KX673248 the following motifs: I) RNA helicase, genome location 566–685; II) 3C cysteine protease, genome location 1340–1378; III) RNA dependent RNA polymerase, genome location 1449–1922; IV) RhV picornavirus capsid, genome location 2009–2123 and V) calicivirus coat protein, genome location 2032–2136 and VI) Waikavirus capsid protein, genome location 2370–2457 (Fig 1b). Only RNA helicase, RNA dependent RNA polymerase, 3C cysteine protease and RhV picornavirus capsid motifs were found in the Brazilian HuV. Their location in the genome of TO-030 are 565–648, 1478–1946, 1367–1405 and 2042–2151, respectively (Fig 1b).

## Phylogenetic analysis

A detailed phylogenetic analysis was performed to determine the relatedness of Brazilian sequences with other Posa and Posa-like viruses detected in distinct hosts. The maximum likelihood tree constructed with viral genomes show that all HuV sequences generated in this study are monophyletic (highlighted in gray in Fig 2). The tree also shows that Posa and Posa-like viruses detected in a certain host are not monophyletic (hosts are indicated by distinct color in the tree). For example, viruses isolated in pigs (indicated in green color) do not share a common ancestor. Likewise, sequences of HuV (indicated in blue) detected recently, form a cluster that includes sequences detected in pigs. Sequences of HuV are distributed in three clades (here named HuV-A, B and C) that likely represent distinct species. In addition, HuV-A is also having two phylo clades (A1 and A2 in the tree). It is important to note that in South America HuV-A, B are circulating. More important is the fact that HuV-A, B were identified in isolated Amerindians in Amazonian regions of Venezuela (sequences in clades indicated by gray rectangles in the tree). Besides, in South America the divergent clades A1 and A2 circulate in Brazil and Venezuela, respectively. Because sequences from distinct countries are intermingled in the tree it is likely that HuV are worldwide spread because. We used the polyprotein region of these genomes to calculated the genetic divergence of sequences (indicated within parenthesis in the branches of the tree) in the clade A1 which is 10%, clade A2 is 12%, clade B is 3% and clade C is 4%. The genetic distance between A1 and A2 is 40%.

## Discussion

We report here the first presence of HuV for the first time in Brazil in five children in the urban and rural areas of North and Northeast states. HuV was initially detected in stool samples in the Netherlands and since then it has been reported in a low frequency in other countries such as Venezuela and Viet Nan [1–4]. There is still limited number of sequences of this virus available in order to explore details of its evolution and health risk to humans. There are just 31 HuV nucleotide sequences deposited in the GenBank database, generated in studies with samples from Brazil (n = 5) Venezuela (n = 8), China (n = 9) Vietnam (n = 4), the Netherlands (n = 3) and Ethiopia (n = 2) [1, 6, 10, 12, 13]. Consequently, information of epidemiological and molecular characteristics is scarce. Despite the limited data, it is known that all HuV sequences already described were detected in fecal samples, of individuals with different clinical statuses such as HIV-1 positive individuals [1], individuals with trachoma symptoms [4], individuals with acute diarrhea and asymptomatic patients [3, 6, 8–10, 18, 39]. Our research was similarly conducted in people with acute gastroenteritis. No other enteric pathogens, except for viruses, were found in Brazilian patients known as HuV. It is important to note that rotavirus, which is likely to be the pathogen involved in the induction of gastroenteritis in these infants was also infected in all individuals in which HuV was found. Also, some authors have proposed that the identification of HuV in feces could be associated with parasitic nematode infection [20, 36]. On the other hand, our analysis shows that all children identified by HuV were not infected with helminths of any kind. Given the epidemiological data (sex, era, year of the collection), there is a variety of current data to date, rendering essential inferences difficult. It should be noted, however, that HuV has been identified both in women and in men, in children and adults [1, 6, 10, 12, 13]. Although we have used a well-proofed protocol aimed to reduce contamination of cellular nucleic acids from bacterial cells the presence of HuV in these children, not necessarily indicate that this virus is infecting humans. One important limitation of our study is the fact that there is no case-control trial, aside from the small number of samples and skewed samples that involve only children with gastroenteritis.

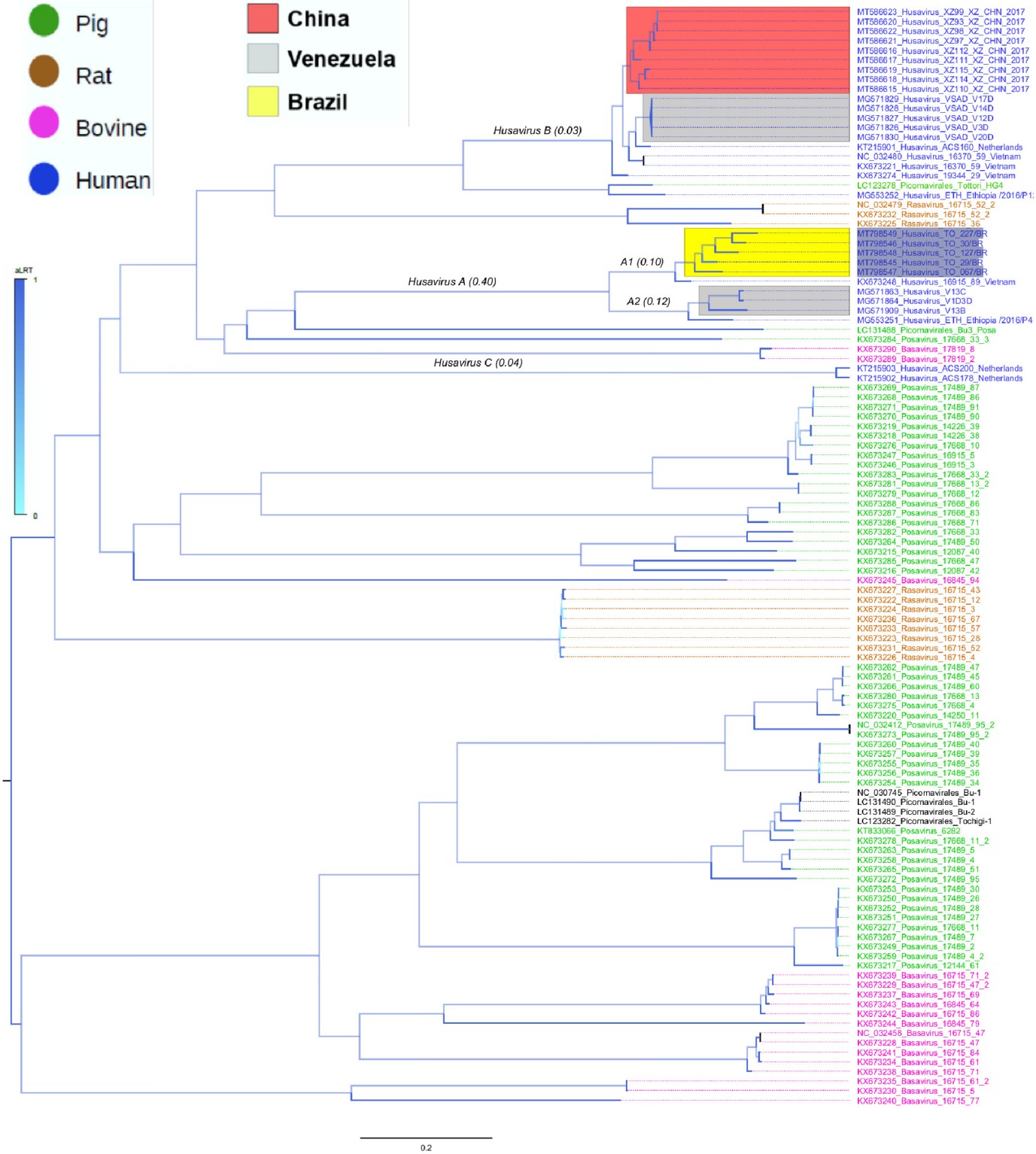

**Fig 2. Maximum likelihood tree of genome of Posa and Posa-like viruses.** Tree was inferred in FastTree using GTR+ gamma correction and the proportion of invariable sites model as selected by the jModelTest software. Branch support was achieved by approximate likelihood ratio test (aLRT) and is shown in a color scale. HuV identified in this study are highlighted in gray. Hosts in which viruses were identified are indicated in colors (green = pig; brown = rat; pink = bovine; blue = human). Colored rectangles indicate clades of HuV sequences from distinct countries (red = China; Gray = Venezuela; Yellow = Brazil). The geographical location of HuV not included in the clades shown by the colored rectangles is indicated in the name of each sequence. This tree was rooted using the poliovirus sequence NC_002058. Numbers within parenthesis above some branches indicate the genetic divergence in phyloclades of HuV.

Perhaps our study's most interesting finding is the diversity of HuV in South America. The Brazilian sequences were almost similar, diverging by less than 10%. Besides, these sequences are connected equally and are extremely similar to one sequence detected in Viet Nan. The lack of geographical segregation of these HuV sequences (clade A1 in the tree) from distant countries suggest that they are globally distributed, while clusters are produced by sequences from the same region. The presence of HuV-A and HuV-B in isolated Amerindians and the very divergent South American presence of HuV-A (clade A1 and clade A2 in the tree) is more important. In conclusion, this is the first study in American Brazil of HuV. Here we demonstrate that helminths, as previously suggested, are unlikely to be the natural hosts of HuV. We also present new data showing that in South America there are divergent HuV lineages. In South America, the existence of many variants (clades) of HuV suggests an intricate evolutionary history, perhaps with repeated events of infection between indigenous people and colonizers.

## Acknowledgments

We thank Coordenação Geral de Laboratórios de Saúde Pública do Departamento de Articulação Estratégica da Secretaria de Vigilância em Saúde do Ministério da Saúde (CGLAB/ DAEVS/SVS-MS), MP Biomedicals do Brasil, Zymo Research Inc. for the donation of reagents for this project. We thank Luciano Monteiro da Silva and Nilton Costa.

## Author Contributions

**Conceptualization:** Ester Cerdeira Sabino, Antonio Charlys da Costa.

**Data curation:** Endrya do Socorro Fôro Ramos, Fabiola Villanova, Flávio Augusto de Pádua Milagres, Vanessa dos Santos Morais, Xutao Deng, Adriana Luchs.

**Formal analysis:** Endrya do Socorro Fôro Ramos, Ulisses Alves Rosa, Geovani de Oliveira Ribeiro, Fabiola Villanova, Adriana Luchs, Élcio Leal, Antonio Charlys da Costa.

**Funding acquisition:** Ester Cerdeira Sabino, Adriana Luchs.

**Investigation:** Antonio Charlys da Costa.

**Methodology:** Vanessa dos Santos Morais.

**Project administration:** Ester Cerdeira Sabino, Eric Delwart.

**Resources:** Flávio Augusto de Pádua Milagres, Rafael Brustulin, Emerson Luiz Lima Araújo.

**Software:** Xutao Deng, Eric Delwart.

**Supervision:** Élcio Leal.

**Writing – original draft:** Endrya do Socorro Fôro Ramos, Ulisses Alves Rosa, Geovani de Oliveira Ribeiro, Élcio Leal, Antonio Charlys da Costa.

**Writing – review & editing:** Ramendra Pati Pandey, V. Samuel Raj, Élcio Leal.

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
