## [Decision Letter · Decision Letter 0]

17 Sep 2020

PONE-D-20-22053

Multiple clades of Husavirus in South America revealed by next generation sequencing of fecal samples.

PLOS ONE

Dear Dr. Leal,

Thank you for submitting your manuscript to PLOS ONE. After careful consideration, we feel that it has merit but does not fully meet PLOS ONE’s publication criteria as it currently stands. Therefore, we invite you to submit a revised version of the manuscript that addresses the points raised during the review process.

Your manuscript has been reviewed by two experts in the field, and they both raised significant issues which need to be addressed before your manuscript can be re-evaluated for publication. Please note both reviewers suggest a careful revision by a English native speaker and proofreading for typos. I encourage you to address the reviewers comments entirely before resubmitting your work. 

We look forward to receiving your revised manuscript.

Kind regards,

Gualtiero Alvisi, PhD

Academic Editor

PLOS ONE

Journal Requirements:

Reviewers' comments:

Reviewer's Responses to Questions

**Comments to the Author**

1. Is the manuscript technically sound, and do the data support the conclusions?

Reviewer #1: Partly

Reviewer #2: Yes

2. Has the statistical analysis been performed appropriately and rigorously? 

Reviewer #1: N/A

Reviewer #2: Yes

3. Have the authors made all data underlying the findings in their manuscript fully available?

Reviewer #1: Yes

Reviewer #2: Yes

4. Is the manuscript presented in an intelligible fashion and written in standard English?

Reviewer #1: Yes

Reviewer #2: Yes

5. Review Comments to the Author

Reviewer #1: The authors describe the identification and phylogenetic analysis of husaviruses for the first time in Brazil. There are some points that need to be clarified.

Lines 37-39 “Detailed molecular characterization indicated that the nucleotide divergence of Brazilian HuVs is less than 11%. On the other hand, the genetic distance between Brazilian sequences and the closest HuV previously described is 18%.”: In what genome region?

Line 104 “New Generation Sequencing (NGS)”: should be “Next Generation Sequencing”.

Line 177 “From the 251 samples analyzed by serology”: How did the authors analyzed by serology? This should be included in the Materials and Methods section.

Table 1: Collection date should be included. “Parvovirus e”, ” Rotavirus e”, ” Torque Teno Virus e”, and ” Adenovirus e” are correct?

Lines 193-195 “We found that the similarity of Brazilian sequences is 92% (in this calculation we excluded sequence TO-029 because it is incomplete). The highest similarity of a Brazilian sequence and the previously described husavirus was 87%,”: In what genome region?

Lines 242-244 “husavirus was initially detected in stool samples in the Netherlands and since then it has been reported in a low frequency in other countries such as Venezuela and Viet Nan [1,2,3,4].”: This sentence is a duplication of Introduction section lines 49-53.

Lines 42-44 “The lack of geographical structure besides the occurrence of distinct lineages in the same region indicates that HuV is an old virus.”, lines 264-266 “More interesting is the presence of a very divergent clade of HuV (clade A2 in the tree) in South American, thus suggesting that husavirus was circulating in the native population before American colonization.”, lines 268-270 “The presence of multiple variants (clades) of husavirus in South America indicate an intricate evolutionary history perhaps with recurrent infection events between native people and colonizers.”: How did the authors estimate that husavirus was circulating in the native population? How did the authors eliminate the possibility of the viruses introduced from other countries or areas?

Figure 2: AB010145_Aichi_virus_genomic may be miss assigned.

Reviewer #2: The authors studied the molecular characterization of husavirus in faecal samples from patients with gastroenteritis in South America. They detected husavirus in only five samples. The authors showed that the viral strains are closely related based on their nucleotide sequence and are closely related to a HuV identified in Viet Nam in 2013. In general, the manuscript sounds good. Although there is nothing novel about the idea nor the techniques which were used to study the molecular characterization of the virus, it is a good study as it is the first on this virus in Brazil. The data of this study may add to global knowledge about the molecular epidemiology of Husavirus. There are crucial points that the authors should elucidate.

6. PLOS authors have the option to publish the peer review history of their article (what does this mean?). If published, this will include your full peer review and any attached files.

Reviewer #1: No

Reviewer #2: No

---

## [Author Response · Author response to Decision Letter 0]

13 Dec 2020

Reviewers' comments:

Reviewer's Responses to Questions

Comments to the Author

Reviewer #1: The authors describe the identification and phylogenetic analysis of husaviruses for the first time in Brazil. There are some points that need to be clarified.

Lines 37-39 “Detailed molecular characterization indicated that the nucleotide divergence of Brazilian HuVs is less than 11%. On the other hand, the genetic distance between Brazilian sequences and the closest HuV previously described is 18%.”: In what genome region?

Resp: to calculate the genetic distance we have used the polyprotein region. This information has been included in the manuscript (line 256, We have used the polyprotein region of these genomes to calculated the genetic divergence of sequences in the clade A1 which is 18%.)

Line 104 “New Generation Sequencing (NGS)”: should be “Next Generation Sequencing”.

Resp: we have changed New to Next (line 128)

Line 177 “From the 251 samples analyzed by serology”: How did the authors analyzed by serology? This should be included in the Materials and Methods section.

Resp: We didn't perform the serological test in these individuals because only fecal samples were available. We did bacterial culture techniques and conventional parasitological tests besides PCR and NGS. This was corrected in the new version of the manuscript (118-128)

Table 1: Collection date should be included. “Parvovirus e”,” Rotavirus e”, ” Torque Teno Virus e”, and ” Adenovirus e” are correct?

Resp: We have included collection dates in table 1 and removed the typos “e”

Lines 193-195 “We found that the similarity of Brazilian sequences is 92% (in this calculation we excluded sequence TO-029 because it is incomplete). The highest similarity of a Brazilian sequence and the previously described husavirus was 87%,”: In what genome region?

Resp: To calculate the nucleotide similarity we have used the polyprotein regions of Husaviruses. We have changed the text accordingly. (line 219: We found that the nucleotide similarity in the polyprotein region of Brazilian sequences is 92%.) 

Lines 242-244 “husavirus was initially detected in stool samples in the Netherlands and since then it has been reported in a low frequency in other countries such as Venezuela and Viet Nan [1,2,3,4].”: This sentence is a duplication of Introduction section lines 49-53.

Resp: In the new version of the manuscript, we have changed the text of the introduction section.

Lines 42-44 “The lack of geographical structure besides the occurrence of distinct lineages in the same region indicates that HuV is an old virus.”, lines 264-266 “More interesting is the presence of a very divergent clade of HuV (clade A2 in the tree) in South American, thus suggesting that husavirus was circulating in the native population before American colonization.”, lines 268-270 “The presence of multiple variants (clades) of husavirus in South America indicate an intricate evolutionary history perhaps with recurrent infection events between native people and colonizers.”: How did the authors estimate that husavirus was circulating in the native population? How did the authors eliminate the possibility of the viruses introduced from other countries or areas?

Resp: We have changed this sentence in the new version of the manuscript,

Figure 2: AB010145_Aichi_virus_genomic may be miss assigned.

Resp:The sequence with the genbank ID AB010145 is indeed a Aichivirus ( isolate A846/88)

Reviewer #2: The authors studied the molecular characterization of husavirus in faecal samples from patients with gastroenteritis in South America. They detected husavirus in only five samples. The authors showed that the viral strains are closely related based on their nucleotide sequence and are closely related to a HuV identified in Viet Nam in 2013. In general, the manuscript sounds good. Although there is nothing novel about the idea nor the techniques which were used to study the molecular characterization of the virus, it is a good study as it is the first on this virus in Brazil. The data of this study may add to global knowledge about the molecular epidemiology of Husavirus. There are crucial points that the authors should elucidate.

Resp: We have updated the sequences used for the comparative phylogenetic analysis and changed the manuscript accordingly.

Review

PLOS One Journal

Multiple clades of Husavirus in South America revealed by next-generation sequencing of faecal samples

The authors studied the molecular characterization of husavirus in faecal samples from patients with gastroenteritis in South America. They detected husavirus in only five samples. The authors showed that the viral strains are closely related based on their nucleotide sequence and are closely related to a HuV identified in Viet Nam in 2013. In general, the manuscript sounds good. Although there is nothing novel about the idea nor the techniques which were used to study the molecular characterization of the virus, it is a good study as it is the first on this virus in Brazil. The data of this study may add to global knowledge about the molecular epidemiology of Husavirus. There are crucial points that the authors should elucidate.

Introduction:

The authors should restructure the introduction as it lacks detailed information in addition to poor English. 

1 How was husavirus discovered in 2015 from human faecal samples? What were the techniques used?Resp: Posa-like viruses, including husavirus, were all identified mostly by NGS from faecal/environmental samples. The introduction has been updated accordingly.

2 The authors stated that the virus could cause a variety of clinical conditions in the host, which host precisely? Human or non-human? Need more resources and references. Resp: We say that members of picornavirales order infect and cause diseases in a variety of hosts. Posa-like viruses, on the other hand, have been detected in faecal and environmental samples linked with animal hosts. However, there is no clear evidence Posa-like viruses cause disease or infect animals. 

3 What is the importance of studying the phylogenetic characterization of the virus? Resp: There are many important features we can obtain by phylogenetic studies of viruses. We can understand and elucidate relatedness and evolutionary aspects of viruses by inferring trees and performing measurements of branch lengths and other key parameters provided by evolutionary models. In our case, we have shown that in South America there are two divergent phyloclades (husavirus A1 and A2). 

4 More references should be included in the introduction related to the clinical importance of the virus. Resp: We have modified several sections of the manuscript and have added new references.

 Study population and specimen collection:

The statement `the protocol used failed to identify possible outbreak` needs more clarification.

Resp: The updated edition of the manuscript omitted this phrase because the experimental design was not aimed at exploring viral outbreaks.

Sample screening:

It is recommended to screen the stool samples first for the presence of common viruses causing gastroenteritis by a routine diagnostic test such as multiplex-Real Time PCR and then perform NGS to identify the known or unknown viruses in the samples. Resp: Samples were screened for viral enteric pathogens (i.e., adenovirus, rotavirus and norovirus), using commercial enzyme immunoassays, such as RotaScreenII ®and AdenoScreen®EIA (Microgen Bioproducts Ltd, 1, Watchmoor Point, Watchmoor Rd, Camberley GU15 3AD, UK).

Results: 

1 The authors mentioned that 251 samples were analyzed by serology or NGS; however, they did not mention that they used serology in the methodology sectiond. Resp:Serology is these individuals was not done. Faecal samples were tested prior to NGS to assess the existence of viruses typically associated with diarrhoea using a commercial immunoassay enzyme. In order to make this detail understandable, we updated the manuscript. 

1 The authors did not remark what is the percentage of the identified HuV in the stool samples?Resp: In 5 samples out of 251 (5/251=1.99 percent) by PCR and NGS, we found Husavirus, which is in the abstract of the new version of the manuscript.

2 Different viruses were detected in the clinical samples using NGS. Were they found as single viruses or co-infection?Resp: Table 1 summarizes the viral species found in individuals whom husavirus was detected. 

3 ‘The highest similarity of the Brazilian sequences and previously described husavirus was 87%.......’. This statement should be restructured to be more understandable.Resp:4. In order to make clear that the similarity between TO-030 and KX673248 is 87 percent, we changed this sentence. 

4 It should be noted to the authors that all identified husaviruses from stool samples belong to clade A1 only, while clade A2 consists of reference husavirus A strains. Resp: Clade A2 sequences were obtained from Amerindians and in an Ethiopian child.

5 The authors have just included seven human husavruses to perform the phylogenetic analysis. Where are the remaining 10 human HuV nucleotides sequences that are deposited in the GeneBank database? Including all the human 17 HuV nucleotides sequences will give a better image on the relatedness of the detected strains in this study to the other strains detected in other countries .Resp: We included 31 Husavirus sequences in this version of the manuscript and this gave us more information about the relatedness of this virus. 

Discussion:

1 The authors stated ‘the presence of a very divergent clade of HuV (clade A2) in South America, which suggest that husavirus was circulating in the native population’. This statement needs lots of explanations and confirmations. What are the other molecular analysis data that support this conclusion?Resp: In order to make it clear that husavirus is an ancient virus, we have changed the discussion and there is no clear indication that it was in native Amerindians before colonization. 

2 Since the HuV strains detected in this study were found as co-infection with other viruses causing gastroenteritis, how can the authors confirm that HuV able to cause clinical outcome as a sole virus? It may be dependant virus! Resp: We didnt say husavirus is causing gastroenteritis we mention that rotavirus is the likely the pathogen causing disease in these children (lines 298-300).

3 The authors did not discuss the limitations of this study. Resp: There is no case-control trial, aside from the small number of samples and skewed samples that involve only children with gastroenteritis. It is also important to remember that our research was not designed to establish whether husavirus infects human cells, while we demonstrate that helminths are not the source of husaviruses.

---

## [Editor Report · Decision Letter 1]

17 Dec 2020

PONE-D-20-22053R1

Multiple clades of Husavirus in South America revealed by next generation sequencing.

PLOS ONE

Dear Dr. Leal,

Thank you for submitting your manuscript to PLOS ONE. After careful consideration, we feel that it has merit but does not fully meet PLOS ONE’s publication criteria as it currently stands. Therefore, we invite you to submit a revised version of the manuscript that addresses the points raised during the review process.

I have carefully gone through your interesting work but, before submitting it to the Reviewers I need to ask you to throughly check it in terms of typos, grammatical rules and readability. As it stands, the manuscript would be very difficult to be revised, and I therefore need to preserve the time and efforts of our Reviewers.

We look forward to receiving your revised manuscript.

Kind regards,

Gualtiero Alvisi, PhD

Academic Editor

PLOS ONE

---

## [Author Response · Author response to Decision Letter 1]

28 Dec 2020

Reviewers' comments:

Reviewer's Responses to Questions

Comments to the Author

Reviewer #1: The authors describe the identification and phylogenetic analysis of husaviruses for the first time in Brazil. There are some points that need to be clarified.

Lines 37-39 “Detailed molecular characterization indicated that the nucleotide divergence of Brazilian HuVs is less than 11%. On the other hand, the genetic distance between Brazilian sequences and the closest HuV previously described is 18%.”: In what genome region?

Resp: to calculate the genetic distance we have used the polyprotein region. This information has been included in the manuscript (line 256, We have used the polyprotein region of these genomes to calculated the genetic divergence of sequences in the clade A1 which is 18%.)

Line 104 “New Generation Sequencing (NGS)”: should be “Next Generation Sequencing”.

Resp: we have changed New to Next (line 128)

Line 177 “From the 251 samples analyzed by serology”: How did the authors analyzed by serology? This should be included in the Materials and Methods section.

Resp: We didn't perform the serological test in these individuals because only fecal samples were available. We did bacterial culture techniques and conventional parasitological tests besides PCR and NGS. This was corrected in the new version of the manuscript (118-128)

Table 1: Collection date should be included. “Parvovirus e”,” Rotavirus e”, ” Torque Teno Virus e”, and ” Adenovirus e” are correct?

Resp: We have included collection dates in table 1 and removed the typos “e”

Lines 193-195 “We found that the similarity of Brazilian sequences is 92% (in this calculation we excluded sequence TO-029 because it is incomplete). The highest similarity of a Brazilian sequence and the previously described husavirus was 87%,”: In what genome region?

Resp: To calculate the nucleotide similarity we have used the polyprotein regions of Husaviruses. We have changed the text accordingly. (line 219: We found that the nucleotide similarity in the polyprotein region of Brazilian sequences is 92%.) 

Lines 242-244 “husavirus was initially detected in stool samples in the Netherlands and since then it has been reported in a low frequency in other countries such as Venezuela and Viet Nan [1,2,3,4].”: This sentence is a duplication of Introduction section lines 49-53.

Resp: In the new version of the manuscript, we have changed the text of the introduction section.

Lines 42-44 “The lack of geographical structure besides the occurrence of distinct lineages in the same region indicates that HuV is an old virus.”, lines 264-266 “More interesting is the presence of a very divergent clade of HuV (clade A2 in the tree) in South American, thus suggesting that husavirus was circulating in the native population before American colonization.”, lines 268-270 “The presence of multiple variants (clades) of husavirus in South America indicate an intricate evolutionary history perhaps with recurrent infection events between native people and colonizers.”: How did the authors estimate that husavirus was circulating in the native population? How did the authors eliminate the possibility of the viruses introduced from other countries or areas?

Resp: We have changed this sentence in the new version of the manuscript,

Figure 2: AB010145_Aichi_virus_genomic may be miss assigned.

Resp:The sequence with the genbank ID AB010145 is indeed a Aichivirus ( isolate A846/88)

Reviewer #2: The authors studied the molecular characterization of husavirus in faecal samples from patients with gastroenteritis in South America. They detected husavirus in only five samples. The authors showed that the viral strains are closely related based on their nucleotide sequence and are closely related to a HuV identified in Viet Nam in 2013. In general, the manuscript sounds good. Although there is nothing novel about the idea nor the techniques which were used to study the molecular characterization of the virus, it is a good study as it is the first on this virus in Brazil. The data of this study may add to global knowledge about the molecular epidemiology of Husavirus. There are crucial points that the authors should elucidate.

Resp: We have updated the sequences used for the comparative phylogenetic analysis and changed the manuscript accordingly.

Review

PLOS One Journal

Multiple clades of Husavirus in South America revealed by next-generation sequencing of faecal samples

The authors studied the molecular characterization of husavirus in faecal samples from patients with gastroenteritis in South America. They detected husavirus in only five samples. The authors showed that the viral strains are closely related based on their nucleotide sequence and are closely related to a HuV identified in Viet Nam in 2013. In general, the manuscript sounds good. Although there is nothing novel about the idea nor the techniques which were used to study the molecular characterization of the virus, it is a good study as it is the first on this virus in Brazil. The data of this study may add to global knowledge about the molecular epidemiology of Husavirus. There are crucial points that the authors should elucidate.

Introduction:

The authors should restructure the introduction as it lacks detailed information in addition to poor English. 

 1 How was husavirus discovered in 2015 from human faecal samples? What were the techniques used?Resp: Posa-like viruses, including husavirus, were all identified mostly by NGS from faecal/environmental samples. The introduction has been updated accordingly.

 2 The authors stated that the virus could cause a variety of clinical conditions in the host, which host precisely? Human or non-human? Need more resources and references. Resp: We say that members of picornavirales order infect and cause diseases in a variety of hosts. Posa-like viruses, on the other hand, have been detected in faecal and environmental samples linked with animal hosts. However, there is no clear evidence Posa-like viruses cause disease or infect animals. 

 3 What is the importance of studying the phylogenetic characterization of the virus? Resp: There are many important features we can obtain by phylogenetic studies of viruses. We can understand and elucidate relatedness and evolutionary aspects of viruses by inferring trees and performing measurements of branch lengths and other key parameters provided by evolutionary models. In our case, we have shown that in South America there are two divergent phyloclades (husavirus A1 and A2). 

 4 More references should be included in the introduction related to the clinical importance of the virus. Resp: We have modified several sections of the manuscript and have added new references.

 Study population and specimen collection:

 • The statement `the protocol used failed to identify possible outbreak` needs more clarification.

Resp: The updated edition of the manuscript omitted this phrase because the experimental design was not aimed at exploring viral outbreaks.

Sample screening:

 • It is recommended to screen the stool samples first for the presence of common viruses causing gastroenteritis by a routine diagnostic test such as multiplex-Real Time PCR and then perform NGS to identify the known or unknown viruses in the samples. Resp: Samples were screened for viral enteric pathogens (i.e., adenovirus, rotavirus and norovirus), using commercial enzyme immunoassays, such as RotaScreenII ®and AdenoScreen®EIA (Microgen Bioproducts Ltd, 1, Watchmoor Point, Watchmoor Rd, Camberley GU15 3AD, UK).

Results: 

 1 The authors mentioned that 251 samples were analyzed by serology or NGS; however, they did not mention that they used serology in the methodology sectiond. Resp:Serology is these individuals was not done. Faecal samples were tested prior to NGS to assess the existence of viruses typically associated with diarrhoea using a commercial immunoassay enzyme. In order to make this detail understandable, we updated the manuscript. 

 1 The authors did not remark what is the percentage of the identified HuV in the stool samples?Resp: In 5 samples out of 251 (5/251=1.99 percent) by PCR and NGS, we found Husavirus, which is in the abstract of the new version of the manuscript.

 2 Different viruses were detected in the clinical samples using NGS. Were they found as single viruses or co-infection?Resp: Table 1 summarizes the viral species found in individuals whom husavirus was detected. 

 3 ‘The highest similarity of the Brazilian sequences and previously described husavirus was 87%.......’. This statement should be restructured to be more understandable.Resp:4. In order to make clear that the similarity between TO-030 and KX673248 is 87 percent, we changed this sentence. 

 4 It should be noted to the authors that all identified husaviruses from stool samples belong to clade A1 only, while clade A2 consists of reference husavirus A strains. Resp: Clade A2 sequences were obtained from Amerindians and in an Ethiopian child.

 5 The authors have just included seven human husavruses to perform the phylogenetic analysis. Where are the remaining 10 human HuV nucleotides sequences that are deposited in the GeneBank database? Including all the human 17 HuV nucleotides sequences will give a better image on the relatedness of the detected strains in this study to the other strains detected in other countries .Resp: We included 31 Husavirus sequences in this version of the manuscript and this gave us more information about the relatedness of this virus. 

Discussion:

 1 The authors stated ‘the presence of a very divergent clade of HuV (clade A2) in South America, which suggest that husavirus was circulating in the native population’. This statement needs lots of explanations and confirmations. What are the other molecular analysis data that support this conclusion?Resp: In order to make it clear that husavirus is an ancient virus, we have changed the discussion and there is no clear indication that it was in native Amerindians before colonization. 

 2 Since the HuV strains detected in this study were found as co-infection with other viruses causing gastroenteritis, how can the authors confirm that HuV able to cause clinical outcome as a sole virus? It may be dependant virus! Resp: We didn't say husavirus is causing gastroenteritis we mention that rotavirus is the likely the pathogen causing disease in these children (lines 298-300).

 3 The authors did not discuss the limitations of this study. Resp: There is no case-control trial, aside from the small number of samples and skewed samples that involve only children with gastroenteritis. It is also important to remember that our research was not designed to establish whether husavirus infects human cells, while we demonstrate that helminths are not the source of husaviruses.

---

## [Decision Letter · Decision Letter 2]

6 Jan 2021

PONE-D-20-22053R2

Multiple clades of Husavirus in South America revealed by next generation sequencing.

PLOS ONE

Dear Dr. Leal,

Thank you for submitting your manuscript to PLOS ONE. After careful consideration, we feel that it has merit but does not fully meet PLOS ONE’s publication criteria as it currently stands. Therefore, we invite you to submit a revised version of the manuscript that addresses the points raised during the review process.

Ineed one Reviewer has still some metodologial issues clearly stated in his/her report.

Furthermore, the quality of writing still requires improvment. For example there are several points of confusion in the text, such in the Abstract section. Please see a few examples below, which however do not represent a comprehensive list of language corrections I encourage you to apply throughout the text.

Line 33. To recognize unknown causes of viruses – please rephrase.

Line 35. Using culture methods and parasitological tests to classify other enteric pathogens such as bacteria, parasites, and viruses, all samples were also examined. – please change to “all samples were also analyzed using culture methods and parasitological tests to classify other enteric pathogens such as bacteria, parasites, and viruses”

Line 37.In 1.9 % of the samples, we observed HuV in three males and two females with a mean age of 2 year. – please change to “1.9 % of the samples samples tested positive for HuV, for a total of 5 positive children, with with a mean age of 2 year, with three males and two females.”

Line 39. The genetic gap between Brazilian sequences and the closest HuV mentioned previously, on the other hand, is 18 %. Please change to “The genetic gap between Brazilian sequences and the closest HuV reported previously, on the other hand, is 18 %.”

Furhermore Figures 1 and 2 are appear to have been inverted.

We look forward to receiving your revised manuscript.

Kind regards,

Gualtiero Alvisi, PhD

Academic Editor

PLOS ONE

Reviewers' comments:

Reviewer's Responses to Questions

**Comments to the Author**

1. If the authors have adequately addressed your comments raised in a previous round of review and you feel that this manuscript is now acceptable for publication, you may indicate that here to bypass the “Comments to the Author” section, enter your conflict of interest statement in the “Confidential to Editor” section, and submit your "Accept" recommendation.

Reviewer #1: All comments have been addressed

2. Is the manuscript technically sound, and do the data support the conclusions?

Reviewer #1: Yes

3. Has the statistical analysis been performed appropriately and rigorously? 

Reviewer #1: N/A

4. Have the authors made all data underlying the findings in their manuscript fully available?

Reviewer #1: Yes

5. Is the manuscript presented in an intelligible fashion and written in standard English?

Reviewer #1: Yes

6. Review Comments to the Author

Reviewer #1: The revised manuscript by Élcio Leal et al. has been improved. But some minor revisions are still required.

Lines 114-117 “Samples were screened for viral enteric pathogens (i.e., rotavirus and norovirus), using commercial enzyme immunoassays, such as RotaScreenII® and AdenoScreen®EIA (Microgen Bioproducts Ltd, 1, Watchmoor Point, Watchmoor Rd, Camberley GU15 3AD, UK).”: The authors mentioned “Samples were screened for viral enteric pathogens (i.e., rotavirus and norovirus), using commercial enzyme immunoassays”; however, they used “RotaScreenII® and AdenoScreen®EIA”. How did they screen for “norovirus”?

Line 122 and elsewhere “Husavirus” or “husavirus”: “husavirus” (lines 230, 232, 235, 236, 246, 261, 265, 266, 267, 269, 273, 291, 292, 300, 305, 307, 308, 310, 315, 323, 326, 327, 328, 329, 330, and 331) should be “HuV”.

7. PLOS authors have the option to publish the peer review history of their article (what does this mean?). If published, this will include your full peer review and any attached files.

Reviewer #1: No

---

## [Author Response · Author response to Decision Letter 2]

21 Feb 2021

Dear Gualtiero Alvisi

Academic Editor

PLOS ONE

We are glad to know that our manuscript “Multiple clades of Husavirus in South America revealed by next generation sequencing” has been considered within the scope of the PLOS ONE.

We deeply appreciated the detailed and insightful comments made by the referee. We also would like to thank the efforts of the editorial board for processing our manuscript. All efforts were made to enhance the quality of the manuscript and to allow for its expedient publication. We are sure that this paper is greatly improved after making the modifications suggested by the reviewer.

Sincerely

Elcio Leal

Editor Comments:

Line 33. To recognize unknown causes of viruses – please rephrase.

Resp: We have done the change this paragraph.

Line 35. Using culture methods and parasitological tests to classify other enteric pathogens such as bacteria, parasites, and viruses, all samples were also examined. – please change to “all samples were also analyzed using culture methods and parasitological tests to classify other enteric pathogens such as bacteria, parasites, and viruses”.

Resp: We have done the change this paragraph.

Line 37. In 1.9 % of the samples, we observed HuV in three males and two females with a mean age of 2 year. – please change to “1.9 % of the samples samples tested positive for HuV, for a total of 5 positive children, with with a mean age of 2 year, with three males and two females.”

Resp: We have done the change this paragraph.

Line 39. The genetic gap between Brazilian sequences and the closest HuV mentioned previously, on the other hand, is 18 %. Please change to “The genetic gap between Brazilian sequences and the closest HuV reported previously, on the other hand, is 18 %.”

Resp: We have done the change this paragraph.

Furhermore Figures 1 and 2 are appear to have been inverted.

Resp: We have change this and images now correspond to the figure 1 and figure 2.

Reviewer 1

Lines 114-117 “Samples were screened for viral enteric pathogens (i.e., rotavirus and norovirus), using commercial enzyme immunoassays, such as RotaScreenII® and AdenoScreen®EIA (Microgen Bioproducts Ltd, 1, Watchmoor Point, Watchmoor Rd, Camberley GU15 3AD, UK).”: The authors mentioned “Samples were screened for viral enteric pathogens (i.e., rotavirus and norovirus), using commercial enzyme immunoassays”; however, they used “RotaScreenII® and AdenoScreen®EIA”. How did they screen for “norovirus”?

Resp: We have investigated norovirus by NGS and PCR. Results of this screening were published recently (Arch Virol. 2021 Mar;166(3):905-913. doi: 10.1007/s00705-020-04944-5. Epub 2021 Jan 19.PMID: 33462673).

Line 122 and elsewhere “Husavirus” or “husavirus”: “husavirus” (lines 230, 232, 235, 236, 246, 261, 265, 266, 267, 269, 273, 291, 292, 300, 305, 307, 308, 310, 315, 323, 326, 327, 328, 329, 330, and 331) should be “HuV”.

Resp: We have changed husavirus to HuV.

---

## [Editor Report · Decision Letter 3]

1 Mar 2021

Multiple clades of Husavirus in South America revealed by next generation sequencing.

PONE-D-20-22053R3

Dear Dr. Leal,

We’re pleased to inform you that your manuscript has been judged scientifically suitable for publication and will be formally accepted for publication once it meets all outstanding technical requirements.

Kind regards,

Gualtiero Alvisi, PhD

Academic Editor

PLOS ONE
---

## [Editor Report · Acceptance letter]

9 Mar 2021

PONE-D-20-22053R3 

Multiple clades of Husavirus in South America revealed by next generation sequencing. 

Dear Dr. Leal:

I'm pleased to inform you that your manuscript has been deemed suitable for publication in PLOS ONE. Congratulations! Your manuscript is now with our production department. 

Kind regards, 

on behalf of

Dr Gualtiero Alvisi 

Academic Editor

PLOS ONE